# Effect of Aging Heat Treatment on Wear Behavior and Microstructure Characterization of Newly Developed Al7075+Ti Alloys

**DOI:** 10.3390/ma16124413

**Published:** 2023-06-15

**Authors:** Hamza A. H. Abo Nama, İsmail Esen, Hayrettin Ahlatcı, Volkan Karakurt

**Affiliations:** 1Department of Mechanical Engineering, Karabuk University, Karabuk 78050, Turkey; hamzah11alhabib84@gmail.com; 2Department of Metallurgical and Material Engineering, Karabuk University, Karabuk 78050, Turkey; hahlatci@karabuk.edu.tr; 3Saglam Metal A.Ş., Kocaeli 41420, Turkey; volkan.karakurt@saglammetal.com

**Keywords:** Al7075, titanium, Al-10%Ti, microstructure, hardness, dry wear

## Abstract

In this study, Al7075+0%Ti-, Al7075+2%Ti-, Al7075+4%Ti-, and Al7075+8%Ti-reinforced alloys were prepared by melting processes using Al7075 and Al-10%Ti main alloys. All newly produced alloys were subjected to T6 aging heat treatment and some samples were cold rolled at 5% beforehand. The microstructure, mechanical behavior, and dry-wear behavior of the new alloys were examined. Dry-wear tests of all alloys were carried out at a total sliding distance of 1000 m, at a sliding speed of 0.1 m/s, and under a load of 20 N. In the hardness measured after T6 aging heat treatment, the peak hardness of the Al7075+0%Ti-, Al7075+2%Ti-, Al7075+4%Ti-, and Al7075+8%Ti-reinforced alloys was found to be 105.63, 113.60, 122.44, and 140.41 HB, respectively. The secondary phases formed by the addition of Ti to the Al7075 alloy acted as precipitate-nucleation sites during aging heat treatment, further increasing the peak hardness. Compared to the peak hardness of the unrolled Al7075+0%Ti alloy, the increase in the peak hardness of the unrolled and rolled Al7075+8%Ti-reinforced alloys was 34% and 47%, respectively, and this difference in the increase was due to the change in the dislocation density with cold deformation. According to the dry-wear test results, the wear resistance of the Al7075 alloy increased by 108.5% with a reinforcement of 8% Ti. This result can be attributed to the formation of Al, Mg, and Ti-based oxide films during wear, as well as the precipitation hardening, the secondary hardening with acicular and spherical Al_3_Ti phases, the grain refinement, and solid-solution-hardening mechanisms.

## 1. Introduction

Aluminum (Al) and aluminum alloys have features such as low density, good thermal and electrical conductivity, and resistance to oxidation [1]. Due to their strength and lightness, Al alloys have recently attracted increasing interest in the aerospace, automotive, marine, packaging, construction, and defense industries [2,3]. The usage of pure aluminum is limited due its softness and weak structure. Therefore, pure aluminum is alloyed to improve its mechanical properties. Alloying elements used to improve its mechanical properties should increase yield and tensile strength and should not compromise the strengths of aluminum, such as high ductility and wear resistance. In addition, the alloying elements used should increase the low melting temperature of pure aluminum and improve its high-temperature performance. Al7075 is very popular among aluminum alloys, and 7xxx series Al-Zn-Mg-Cu alloys are widely used in harsh working conditions due to their age hardenability, high fracture toughness, and strength [4,5,6].

In addition, the precipitation process increases the yield strength of the alloy because the precipitates inhibit the dislocation motion in the microstructure of the material. The magnitude of the strengthening effect is naturally related to the chemical composition of the precipitates and the interfacial energies [7]. Mechanical properties of Al7075 alloy can be further improved, in particular by thinning the microstructure through recrystallization heat treatment following heavy plastic deformation [8]. Titanium is a relatively light element that exhibits good wear resistance, a high strength-to-density ratio, and good high-temperature properties. Al-Ti alloys are considered one of the most important material groups of the future because they have the advantages of high elongation ability and lightness for the strength ratio. Several researchers are working on microstructural improvements such as grain refinement to further improve these properties of Al7075 alloy [9]. For example, Kumar et al. [10] prepared four different Al7075 alloy-matrix composites filled with Ti metal powder with a filler content between 0 and 2 wt% using the vacuum-mix-casting method. They compared the dry-wear performance of these composites and reported that the 2% Ti metal-powder-filled Al7075 composite had the lowest specific wear rate. Rajan et al. [11] studied Al7075 alloy composites reinforced with titanium diboride particles and observed that the microhardness and tensile strength of TiB2-doped aluminum-matrix composites increased with higher particle content. Surya et al. [12] studied the effects of SiC amount on the wear behavior and mechanical properties of SiC-reinforced Al7075 composites in powder metallurgy. The authors reported that the microstructure and mechanical properties of SiC-reinforced Al7075 composites deteriorated and wear resistance decreased depending on the increase in percent SiC reinforcement by weight.

Gajakosh et al. [13] produced hybrid metal-matrix composites of Al7075 + titanium dioxide + graphite by the mixing process and hot rolling. They reported that the reinforcement was better distributed in the hot-rolled composite specimens, and the reinforced and rolled composites showed greater increases in hardness and tensile strength compared to unrolled and unreinforced composites. Qin et al. [14] produced Ti/Al layered composites using hot-pressing processes and reported that a good bonding interface was formed between the layers forming the composite and that small gaps and intermetallic compounds were not formed at the interfaces. Partheeban et al. [15] prepared Al6061-10TiB2, Al6061-10TiB2-2Gr, and Al6061-10TiB2-4Gr nano-hybrid composites using the powder-metallurgy method and examined their wear behavior to find that TiB2 and nanographene reinforcement increased the hardness of the Al6061 matrix. They observed that the wear resistance and friction coefficient of the hybrid composites were better than the wear resistance and friction coefficient of the Al6061-10TiB2 composite. Wu et al. [16] studied the microstructure and mechanical properties of an Al-Ti alloy that solidifies rapidly after heat treatment. They concluded that the Al_3_Ti particles were homogeneously dispersed in the Al matrix and that the hardness and yield strength improved as the volume fraction of the Al_3_Ti particles increased. Wang and Kao [17] and Guoaxion et al. [18] analyzed the formation process of Al_3_Ti by rapid solidification and mechanical alloying, and reported high yield strength in spherical Al_3_Ti grain samples. Liu et al. [19] reported that Al_3_Ti nucleation at melting temperature in Al7075 matrix improved its mechanical properties and microstructure. Rong et al. [20] studied the impact of the chemical mixture K_2_TiF_6_-CaF_2_-LiCl added to Al6351 alloy at 720 °C on the sizes and shapes of intermetallic compounds such as Mg_2_Si, Al_3_Ti, and Al_3_TiSi_0.22_. Wang et al. [21] studied the effect of Al_3_Ti formation by aluminothermic reduction of hexafluorotitanate (K_2_TiF_6_) in situ. Shaeri et al. [22] examined the impact of the same application on the aging heat treatment of Al7075 by applying ECAP (Equal Channel Angle Pressing) to Al7075 and reported that ECAP treatment improved the mechanical properties of Al7075. Ramesh et al. [23] and Wang and Reddy [24] studied the wear behavior of Al6061 alloy and Al6061+TiO_2_ composites and found that the wear resistance of Al6061+TiO_2_ composites was higher than that of Al6061 alloy. Radhika et al. [25] added 6% by weight of Ti-based TiB_2_, TiO_2_, and TiC particles to A359 alloy and reported that the TiB_2_ composite exhibited better hardness and wear resistance in reinforced homogeneous composites. It has also been reported in the literature that applying the cold-rolling process to Al7705 alloy with different parameters increases the hardness and mechanical properties of Al7075 alloy [26,27]. Reddy et al. [28] studied the wear behavior of Al6063+TiC-reinforced alloy at 10, 20, and 30 N and reported that the percentage loss at 30 N load weight was higher than the loss at the other two loads, which was due to increased particle reinforcement. Niranjan et al. [29] performed the pin-on-disc wear test of the Al/TiB_2_ composites they produced and observed that the wear resistance increased with higher TiB_2_ reinforcement.

The addition of Ti to Al7075 alloy by the casting process with up to 2% titanium particles has been examined in the literature [10]. In this study, new Al7075+2–8%Ti alloys, which have not been studied before in the literature, were prepared by the melting process using Al7075 alloy and an Al-10%Ti master alloy. The Al7075+0%Ti, Al7075+2%Ti, Al7075+4%Ti, and Al7075+8%Ti alloys produced by the melting process were homogenized for 5 h at 490 °C, and then the solid solution was annealed for 2 h at 550 °C and finally subjected to a T6 aging heat treatment for 96 h to improve its mechanical properties. In addition, some samples were rolled at a ratio of 5% to study the impact of the applied plastic strain on the mechanical properties of the alloys. The microstructure characterization, hardness tests, and dry-wear behavior of the developed alloys were examined in detail and the results are presented. According to the results obtained in this study, it is claimed that the newly produced alloys can have new alternative uses in critical applications in the aerospace and automotive industries.

## 2. Materials and Methods

In this study, Al7075+0%Ti-, Al7075+2%Ti-, Al7075+4%Ti-, and Al7075+8%Ti-reinforced alloys were produced by the casting method in an induction-melting furnace. The titanium content of the Al-10%Ti master alloy was employed to transform the titanium content of the Al7075 alloys in the given ratio. Accordingly, 200 g of master alloy containing 20 g of titanium were used for a total of 1000 g of Al7075+2%Ti composition, as given in Table 1. Since the Al content in the master Al7075+10%Ti alloy was expected to change the composition of the analyzed Al7075 alloy and partial Mg content would be oxidized during the casting process, an average of 9.6 g Mg calculated in Table 1 was added to the molten mixture. As shown in Table 2, the initial weight of the Al7075 alloy was melted in a graphite crucible in a gas-shielded induction furnace. After increasing the mixing temperature to [4] 780 °C, Al-10%Ti master alloys were placed in the melt in the form of ingots, where they were kept for half an hour and mechanically stirred with a graphite rod to form a homogeneous mixture. After the melted mixture was fully homogenized, Mg additions were completed, and the researchers waited for 5 min. Since magnesium oxidizes quickly, Mg was added last to prevent a reduction in the concentration ratio of the magnesium in the mixture. Finally, alloy materials were obtained by casting the compositions in metal molds with a diameter of 32 mm and a height of 200 mm.

The chemical compositions of the new alloys produced by adding the Al-10%Ti master alloy are given in Table 2. Accordingly, the proportions of Si, Fe, Cu, Mn, Cr, and Zn alloying elements in the alloy (Table 2) decreased due the higher Ti amount added with the master alloy.

Homogenization procedures of all casted alloys were performed in an atmospheric controlled furnace at 490 °C for 5 h, as recommended in the literature [30,31]. After homogenization, the analyzed alloys were kept at 500 °C for 2 h for solid-solution formation before being rapidly cooled by water quenching. After this step, alloys were rolled at 10 rpm for 5% cold work before aging. Roll spacing based on the rolling-deformation ratio was calculated using Equation (1) below.
(1)$$h_{1}=\frac{h_{0}}{e^{\varepsilon }}$$
where $h_{0}$ is the initial thickness, $h_{1}$ is the final thickness, ε is the deformation ratio, and e=2.7182.

Then, the rolled and unrolled alloys were subjected to T6 aging heat treatment for up to 96 h at 120 °C, which is recommended as the most appropriate aging temperature in the literature, in order to increase their strength [32]. For the metallographic examination of the rolled and unrolled alloys, the alloys were first sanded with sandpaper in the range of 600, 800, 1000, 1200, and 2500 grids, and surface cleaning was completed by polishing with 1 μm alumina paste. Then, the alloys were etched in the HF stiction solution (1 mL HF (48%, 200 mL H_2_O) by swab at 27 °C for 25 s to reveal their microstructure details. Microstructural characterization of the etched alloys was analyzed by an optical-light microscope (Nikon inverted metal microscope (Nikon, Melville, NY, USA) with Clemex image-analysis software: Clemex vision lite), SEM (Carl Zeiss Ultra Plus Gemini Fesem (Zeiss, Oberkochen, Germany)), and XRD (Rigaku Ultima IV brand X-ray Diffraction Spectrometer (Rigaku, Woodlands, TX, USA)).

The hardness measurements of the alloys were conducted with a Brinell hardness device using a 2.5 mm ball and a load of 187.5 N. The abrasion tests of the produced composites were carried out in a dry environment with the reciprocating tribometer test device (Figure 1). The wear tests were carried out at a constant sliding speed of 0.1 m/s under a load of 20 N and a total sliding length of 1000 m. An AISI 52,100 high-quality-alloy 6 mm-diameter steel ball was used as the counter surface. Under the applied load of 20 N, the reciprocal motion of the 6 mm-diameter steel ball scratched the surface of the abraded alloy with high contact pressure, separating the reinforcements in its microstructure. Reciprocating sliding-friction and wear tests can provide valuable information on the wear resistance, coefficient of friction, and wear mechanism of composites. The friction coefficients of the alloys were calculated using the resistive forces measured at 200 m by a load cell during the dry-friction test. The weight losses of the alloys were measured at intervals of 200 m with a balance with an accuracy of 0.0001 g. Wear rates were calculated in g/mN using weight loss per unit slip distance and load. Wear-rate values were evaluated as the slope of the lines in the weight loss vs. sliding distance graphs. After each wear test, the worn surfaces of the alloys were examined in detail with SEM and EDX analyses and the wear mechanisms of the alloys were determined.

## 3. Results

### 3.1. XRD Analysis

Figure 2 shows the XRD results of the Al7075+0%Ti-, Al7075+2%Ti-, Al7075+4%Ti-, and Al7075+8%Ti-reinforced alloys. The XRD result of the Al7075+0%Ti alloy (Figure 2a) provides the intensity counts of the α-(Al) matrix, Al_2_Cu, Al_2_CuMg, Al_13_Fe, Mg_2_Si, and MgZn_2_ phases based on the diffraction peaks. According to Table 2, binary and triple phases containing Al, Zn, Cu, Fe, Si, and Mg elements were formed in the examined alloys. As seen in Figure 2b–d, the Al_3_Ti phase in addition to the above phases was formed in the investigated alloys, reinforcing up to 8% Ti. In Al-Ti alloys, when the melt temperature in the metal mold approached 665 °C, the Al_3_Ti phase was produced from the liquid and the Al_3_Ti phase remained in the α-(Al) matrix grains at room temperature [19].

### 3.2. Microstructural Characterization

Figure 3 shows the light-microscopy microstructure images of the examined alloys. Grain size was calculated according to the ASTM standard (ASTM E112) [33] using a software (Clemex vision lite) image-processing program that took the average value of five measurements. The average grain sizes of the unrolled Al7075+0%Ti-reinforced alloy, Al7075+2%Ti-reinforced alloy, Al7075+4%Ti-reinforced alloy, and Al7075+8%Ti-reinforced alloy were calculated as 55 µm, 28 µm, 20 µm, and 12 µm, respectively, and the average grain sizes of the rolled Al7075+0%Ti-reinforced alloy, Al7075+2%Ti-reinforced alloy, Al7075+4%Ti-reinforced alloy, and Al7075+8%Ti-reinforced alloy were found to be 48 µm, 22 µm, 15 µm, and 9 µm, respectively. Nucleation began around the initial Al_3_Ti solid particles created during solidification [19]. As the Ti concentration of the alloy increased, more Al_3_Ti solids began to form in the melt, more fine grains appeared, and the grains became further refined. The grain-size refinement by the rolling process can be attributed to the dislocation density formed.

It was determined that Al_3_Ti intermetallics were present within grains and at grain boundaries in the microstructure images presented in Figure 3c–g and increased with higher titanium content in the alloy (see SEM and EDX analyses in Figure 4 and Figure 5, respectively, for details). However, some of the Al_3_Ti intermetallics in the Al7075+4%Ti- and Al7075+8%Ti-reinforced alloys were needle-like, whereas others were spherical. It was observed that both the spherical and needle-like Al_3_Ti intermetallics were more common in the 8% titanium-added alloy compared to the 4% titanium-added alloy. Factors such as grain size, dislocation density after deformation, and needle-like and spherical intermetallics are believed to affect the mechanical behavior of the examined alloys. In the rolled alloys (Figure 3d,f,h), pointed (needle-like) Al_3_Ti was observed instead of spherical Al_3_Ti.

Figure 4 shows the SEM image of the rolled Al7075+0%Ti alloy at 1 k× magnification. According to the results of the elemental-response-spectrum analysis reported in Table 3, the regions marked with 1 and 2 in Figure 4 contained on average 2.96% Mg, 85.31% Al, 0.45% Si, 3.81% Cu, and 7.79% Zn elements, whereas the region with 3 generally included Zn-rich compositions. The microstructure of the titanium-free alloy was mainly composed of the α-(Al) matrix, Al_2_Cu and Al_2_CuMg phases rich in the Cu element, and a MgZn_2_ phase rich in the Zn element.

Figure 5 shows the SEM image of the rolled Al7075+8%Ti alloy at 1 k×. magnification. According to the elemental-response-spectrum analysis presented in Table 4, an average of 88.74% Al, 1.79% Mg, 0.28% Ti, 6.54% Cu, and 2.65% Zn were found in the region indicated by 1 in Figure 5, which shows the main Al matrix. It reveals that the phases with spherical and needle-like morphology, numbered 2 and 4 in Figure 5, respectively, were composed of Al_3_Ti intermetallic. According to the diffusion-phase-transition theory, the coarsening of the generated Al_3_Ti phases in the 8% Ti-reinforced alloy is related to increasing phase-growth rate and solute concentration: Phase growth is accelerated by solute concentration [34]. Therefore, aggregation and growth of Al_3_Ti particles were observed in the Al7075+8%Ti alloy. The area in the bright region indicated by number 3 in Figure 5 contained Mg, Al, Si, Fe, and Cu elements in the ratios of 0.01%, 11.27%, 0.14%, 2.62%, and 85.96%, respectively. The rolled Al7075+8%Ti alloy, based on this composition, could produce phases defined as Al_2_Cu and Al_2_CuMg.

The mass densities of the sample were defined using the Archimedes principle, and they, were 2.813392, 2.74818, 2.69198, and 2.57958 (g/cm^3^) for the Al7075+0%Ti-reinforced alloy, Al7075+2%Ti-reinforced alloy, Al7075+4%Ti-reinforced alloy, and Al7075+8%Ti-reinforced alloy, respectively.

### 3.3. Hardness-Test Results

The maximum increase in hardness of alloys rolled and unrolled for hardening usually occurs during the 48 h aging period. It was seen that %Ti reinforcement in alloys caused an increase in the hardness of the material. Maximum hardness values of the unrolled (Figure 6a) and rolled (Figure 6b) Al7075+0%Ti alloys in the 48 h T6 aging heat treatment were 95.13 HB and 105.63 HB, respectively. Maximum hardness values of unrolled and rolled Al7075+8%Ti-reinforced alloys in 48 h T6 aging heat treatment were 127.87 HB and 140.11 HB, respectively. Accordingly, the hardness of each 5% rolled alloy was measured to be approximately 10% higher on average compared to that of the unrolled alloys. Among the examined materials, the highest hardness value was measured in the rolled alloy with 8% Ti addition, and the increase in the hardness of the rolled Al7075+8%Ti-reinforced alloy was 47% higher than that of the unrolled Al7075+0%Ti alloy. The improvement in strength can be attributed to the inhibition of the dislocation movement by the Al_3_Ti intermetallics and fine grains due to the grain-refining effect of the Ti element as well as the strengthening of the solid solution. Although the Al7075+0%Ti alloy generated the phases Al_2_Cu, Al_2_CuMg, Al_13_Fe_4_, Mg_2_Si, and MgZn_2_, the microstructure of the alloys including up to 8% Ti additionally showed (Figure 3 and Figure 5) the appearance of spherical and needle-like Al_3_Ti phases with increasing growth rate.

### 3.4. Wear-Test Results

Figure 7 shows the dry-wear-test weight loss with a sliding distance of up to 1000 m for the examined Al7075+0%Ti-, Al7075+2%Ti-, Al7075+4%Ti-, and Al7075+8%Ti-reinforced alloys in unrolled and rolled conditions. Unrolled Al7075+0%Ti alloy suffered from the greatest weight loss over a sliding distance of 1000 m with 0.0510 g, as shown in Figure 7a. The weight loss at the same distance was 0.0291 g for the unrolled Al7075+2%Ti-reinforced alloy, whereas that of the unrolled Al7075+4%Ti-reinforced alloy was 0.0257 g. The lowest weight loss among the unrolled alloys was recorded for the unrolled Al7075+8%Ti-reinforced alloy with 0.0188 g. The weight loss of the unrolled Al7075 alloy reinforced with 8% Ti was 63.13% less than that of the unrolled Al7075 alloy without Ti addition. Figure 7b displays the weight losses for the 5% rolled Al7075+0%Ti-, Al7075+2%Ti-, Al7075+4%Ti-, and Al7075+8%Ti-reinforced alloys during the dry-wear test. At 1000 m sliding distance, the weight loss in the rolled Al7075+0%Ti-, Al7075+2%Ti-, Al7075+4%Ti-, and Al7075+8%Ti-reinforced alloys was 0.0353 g, 0.0247 g, 0.0223 g, and 0.0179 g, respectively. Comparing the wear results of the whole rolled alloys with the unrolled Al7075+0%Ti alloy, it was found that the weight loss of the examined rolled alloys showed an improvement of 64.90% with 8% Ti reinforcement. Compared to the wear behavior of the unrolled Al7075+0%Ti alloy, the fact that the wear-behavior improvement rates of the unrolled and rolled Al7075+8%Ti alloys were close to each other means that the effect of rolling on the wear resistance of high-Ti-containing alloys was reduced. This can be attributed to the presence of a common needle-like Al_3_Ti phase in the rolled Al7075+8%Ti-reinforced alloy.

The wear rates of the unrolled and rolled alloys calculated in the unit of g/m from the mean slopes of the weight-loss–slip-distance curves presented in Figure 7 are given in Figure 8. Despite the fact that the wear rates of the unrolled and rolled alloys were reduced when Ti was added (Figure 7), the difference between their wear rates converged at each Ti content, as shown in Figure 8. Whereas the change in the wear rates for the unrolled and rolled Al7075+0%Ti alloys was 31%, it decreased to 6% for the unrolled and rolled Al7075+8%Ti alloys. This result was consistent with the change in the hardness value of the unrolled and rolled Al7075+8%Ti alloys. The variation between the hardness scores of the unrolled and rolled Al7075+8%Ti alloys was approximately 9%. The increase in wear resistance with Ti addition can be attributed to the formation of an oxide film on the worn surface of the alloy with the heat generated during the friction, together with the reasons causing the increase in strength (such as the inhibition of the dislocation movement of the Al_3_Ti intermetallics, the grain-refinement effect of the Ti element, and the solid solution strengthening). Figure 9, Figure 10, Figure 11 and Figure 12 show the appearance of the wear surfaces and the EDX-MAP analyses of the examined alloys.

Figure 9a shows the SEM image of the worn surface of the unrolled Al7075+0%Ti alloy at 1 k× magnification and Figure 9b shows the EDX-MAP analysis results for the elements Al, Mg, and O. As can be seen in Figure 9a, this alloy displayed adhesive- and especially abrasive-wear mechanisms. In Figure 9a, the area marked A shows the abrasive-wear mechanism and the area marked B shows the adhesive-wear mechanism. In addition, the adhesive-wear mechanism in B included some oxide layers. Based on the results of the EDX-MAP analysis of the elements Al, Mg, and O given in Figure 9b, it was observed that stratified alumina (Al_2_O_3_) and magnesia (MgO) layers were formed on the adhesive-wear surfaces.

Figure 10a shows the SEM image of the worn surface of the rolled Al7075+0%Ti alloy at 1 k×. magnification and Figure 10b shows the EDX-MAP analysis results for the elements Al, Mg, and O. As seen in Figure 10a, the wear mechanism of this rolled alloy was also adhesive and abrasive, with adhesive wear being dominant. In Figure 10a, the areas indicated by A show the abrasive-wear mechanism, whereas the area indicated by B represents the adhesive-wear mechanism. In addition, the adhesive-wear mechanism in B showed oxide-layering characterization, as in the rolled alloy. Based on the results of the EDX-MAP analysis of the elements Al, Mg, and O given in Figure 10b, it was observed that accumulated alumina (Al_2_O_3_) and magnesia (MgO) layers were formed on the adhesive-wear surfaces. Some layers were newly formed, whereas others were considered stratified. When the EDX images were examined, it was observed that the Al_2_O_3_ layer on the surface was much larger than the MgO layer. Compared to the unrolled Al7075+0%Ti alloy in Figure 9, the oxidation was slightly higher in the rolled alloy. The MgO content in the unrolled Al7075+0%Ti alloy was higher than in the rolled alloy.

Figure 11a shows the SEM image of the worn surface of the unrolled Al7075+8%Ti alloy at 1 k× magnification and Figure 11b shows the EDX-MAP analysis results for the elements Al, Mg, Ti, Si, Zn, and O. In Figure 11a, region A shows the abrasive-wear mechanism, Region B shows the adhesive-wear mechanism, and Region C shows the formation of abrasive and pre-layer cracks (delamination). Different types of oxide layers were found in the adhesive-wear zones in the area labelled B in Figure 11a. Based on the results of the EDX-MAP analysis of the elements indicated in Figure 11b, these were assumed to be mainly Al_2_O_3_ oxide layers. When the types and amounts of the oxide layers on the surface were assessed, it was seen that the most widespread ones were Al_2_O_3_ layers, followed by MgO and TiO_2_ layers.

Figure 12a shows the SEM image of the worn surface of the rolled Al7075+8%Ti alloy at 1 k×. magnification and Figure 12b shows the EDX-MAP analysis results of the elements Al, Mg, Ti, Si, Zn, and O. Figure 12a shows the abrasive-wear mechanism in the area marked A, the adhesive-wear mechanism in the area marked B, and the abrasive coating in the area marked C. In addition, some oxide particles of the SiO_2_ and ZnO were observed on the surface. The oxidation was very high in this rolled alloy compared to the unrolled Al7075+8%Ti alloy shown in Figure 11a. In addition, more oxide layers were formed here compared to non-titanium alloys. In the wear behavior of the Al7075+0%Ti alloys without the addition of titanium, shown in Figure 9 and Figure 10, more oxidation was observed in the rolled compared to the unrolled alloy. When these results are evaluated together, it can be argued that rolling process increased oxidation on the worn surface of the alloys.

For comparison, Figure 13 shows the average coefficients of friction of the rolled and unrolled alloys calculated with the friction-force measurement, conducted every 200 m. Accordingly, the average coefficient of friction was calculated as 0.0495 for the unrolled Al7075+0%Ti alloy and 0.0401 for the rolled Al7075+0%Ti alloy.

Accordingly, the ratio of the friction coefficients of the rolled- and unrolled-alloy samples was calculated as 1.23. The friction coefficients for the rolled and unrolled Al7075+2%Ti alloys were calculated as 0.0282 and 0.04264, respectively. Thus, the friction-coefficient ratio between unrolled and rolled samples was 1.51. The friction coefficients of the rolled Al7075+4%Ti and unrolled Al7075+4%Ti alloys were measured as 0.0194 and 0.0321, respectively, and the rate of change was calculated as 1.65 accordingly. The friction coefficients of the unrolled Al7075+8%Ti and rolled Al7075+8%Ti alloys were measured as 0.0229 and 0.0186, respectively, and the rate of change was calculated as 1.23 accordingly. The increase in the oxygen content of the oxide film formed on the worn surface led to an increase in wear resistance and a decrease in the friction coefficient.

## 4. Conclusions

In this study, new Al7075+0%Ti, Al7075+2%Ti, Al7075+4%Ti, and Al7075+8%Ti alloys were produced, and some alloys were rolled at a rate of 5% to show the impact of cold rolling. All alloys were exposed to the T6 aging heat treatment for 96 h. The microstructure characterization, hardness analysis, and dry-wear behavior of the newly developed alloys were examined in detail, the results of which are presented below:According to ASTM E112 [33], the average grain sizes of the unrolled Al7075+0%Ti, Al7075+2%Ti, Al7075+4%Ti, and Al7075+8%Ti alloys were determined to be 55, 28, 20, and 12 µm, respectively. On the other hand, the average grain sizes of the rolled Al7075+0%Ti, Al7075+2%Ti, Al7075+4%Ti, and Al7075+8%Ti alloys were determined to be 48, 22, 15, and 9 µm, respectively. It was observed that the Al_3_Ti intermetallic phases increased with better titanium reinforcement in the microstructure. It was also shown that the intermetallic Al_3_Ti phases were generally acicular in the rolled alloys, whereas they were usually spherical in the unrolled alloys.The peak hardness values of the unrolled Al7075+0%Ti and rolled Al7075+0%Ti alloys after 48 h T6 aging heat treatment increased by 11%, whereas the same values for the unrolled Al7075+8%Ti- and rolled Al7075+8%Ti-reinforced alloys increased by 9.4%. On the other hand, the increased rates of hardness of the unrolled and rolled Al7075 alloys with Ti reinforcement of up to 8% were close to each other, at 33%.The wear rates of the examined Al7075 alloys were reduced by approximately 50% when 8% Ti was added. The difference in wear rates between unrolled and rolled Al7075+0%Ti alloys was 31%, whereas it was only 6% for the unrolled and rolled Al7075+8%Ti alloys. This finding is in agreement with the difference in hardness between the examined unrolled and rolled alloys.It was observed that the wear behaviors of the Al7075+0%Ti- and Al7075+8%Ti-reinforced alloys showed higher oxidation under rolled conditions compared to unrolled conditions. Based on these findings and the friction-coefficient data, it can be said that rolling and reinforcement with up to 8% Ti increases surface oxidation during wear, thus protecting the surface and reducing the coefficient of friction.

## Figures and Tables

**Figure 1 materials-16-04413-f001:**
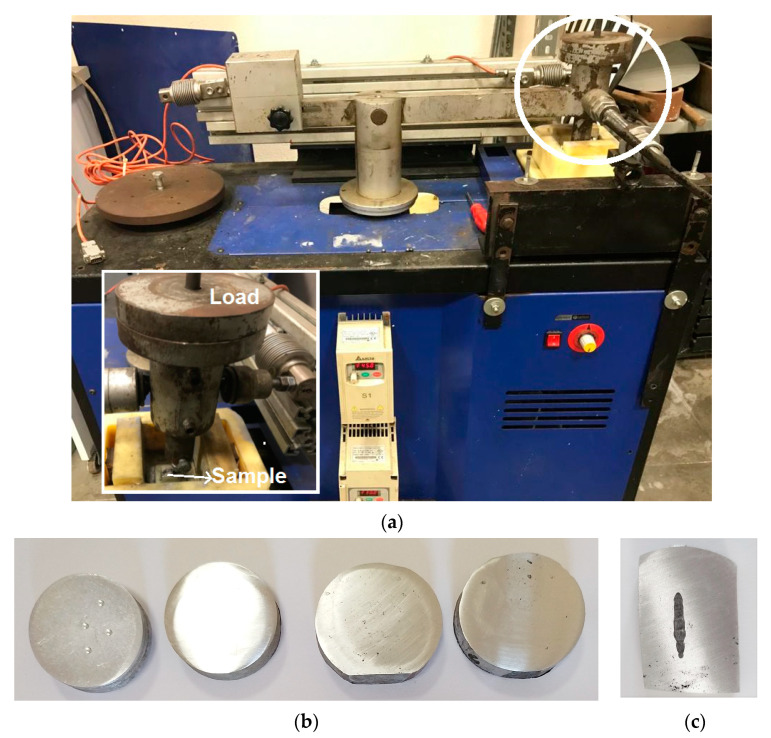
(**a**) The reciprocating wear device; (**b**) some of the produced samples, including Al7075+0%Ti, Al7075+2%Ti, Al7075+4%Ti, and Al7075+8%Ti, from left to right; (**c**) the wear-test sample for Al7075+4%Ti.

**Figure 2 materials-16-04413-f002:**
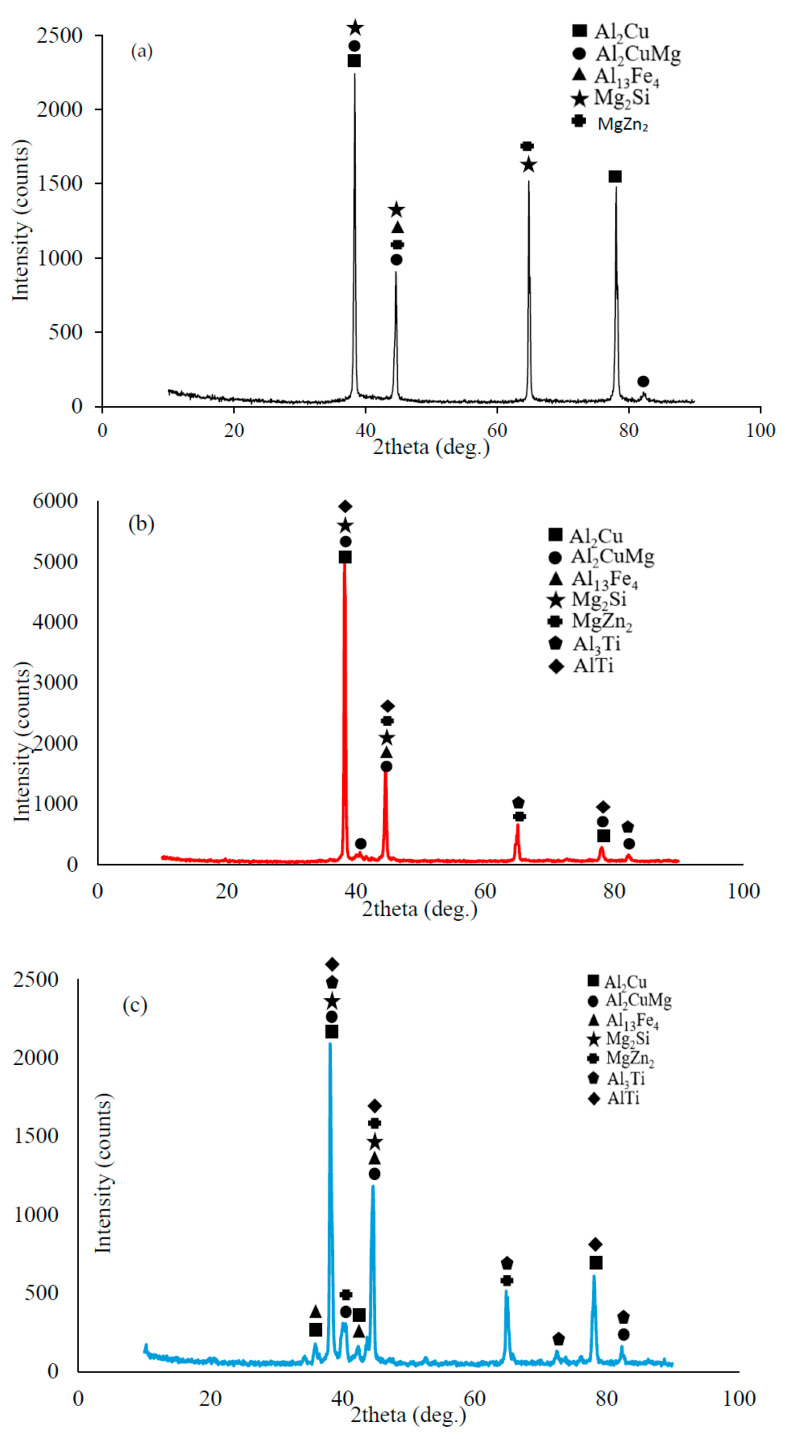
XRD-response spectrum results of the (**a**) Al7075+0%Ti-, (**b**) Al7075+2%Ti-, (**c**) Al7075+4%Ti-, (**d**) Al7075+8%Ti-reinforced alloys.

**Figure 3 materials-16-04413-f003:**
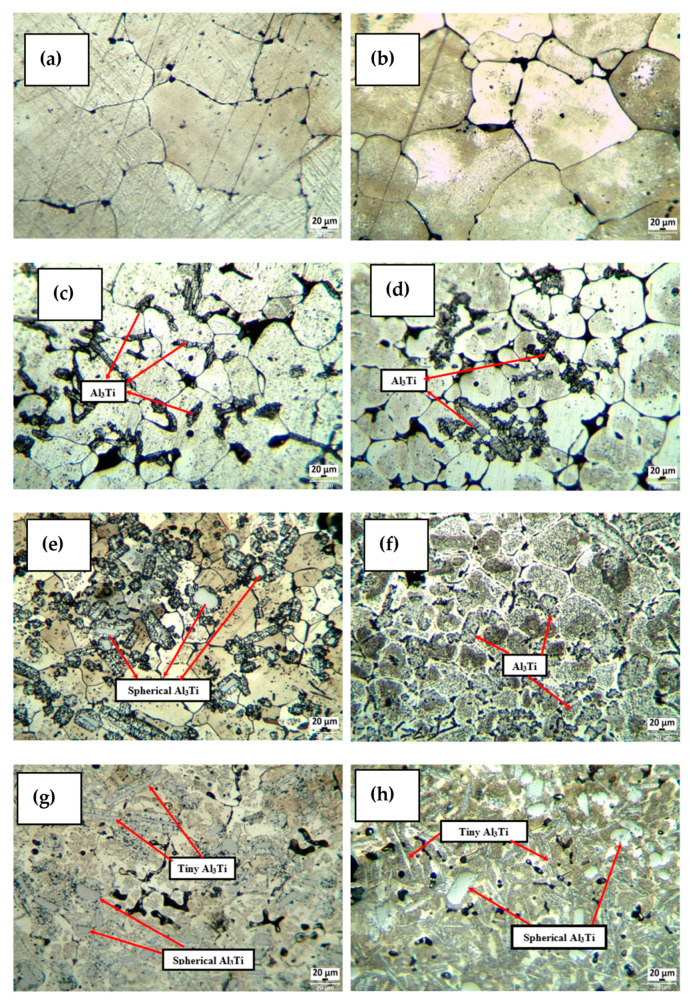
Light-microscopy images of the unrolled and rolled alloys: (**a**) unrolled Al7075+%0Ti alloy; (**b**) rolled Al7075+%0Ti alloy; (**c**) unrolled Al7075+2%Ti alloy; (**d**) rolled Al7075+2%Ti alloy; (**e**) unrolled Al7075+4%Ti alloy; (**f**) rolled Al7075+4%Ti alloy; (**g**) unrolled Al7075+8%Ti alloy; (**h**) rolled Al7075+8%Ti alloy.

**Figure 4 materials-16-04413-f004:**
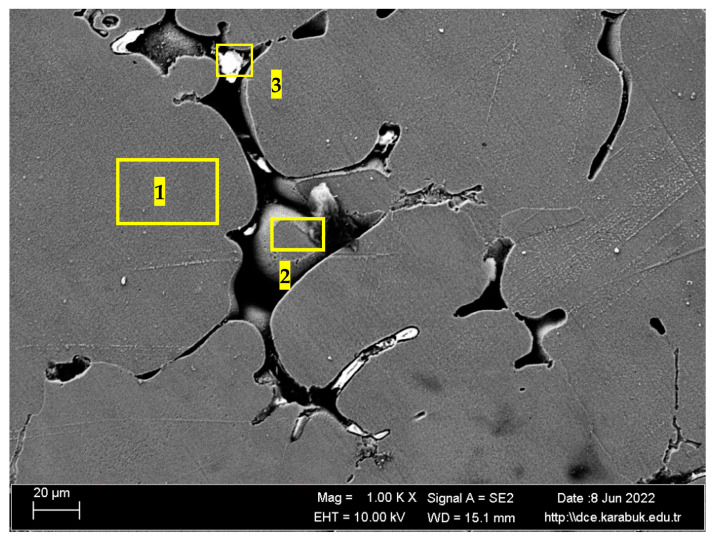
SEM image of the rolled Al7075+0%Ti alloy at 1 k×.

**Figure 5 materials-16-04413-f005:**
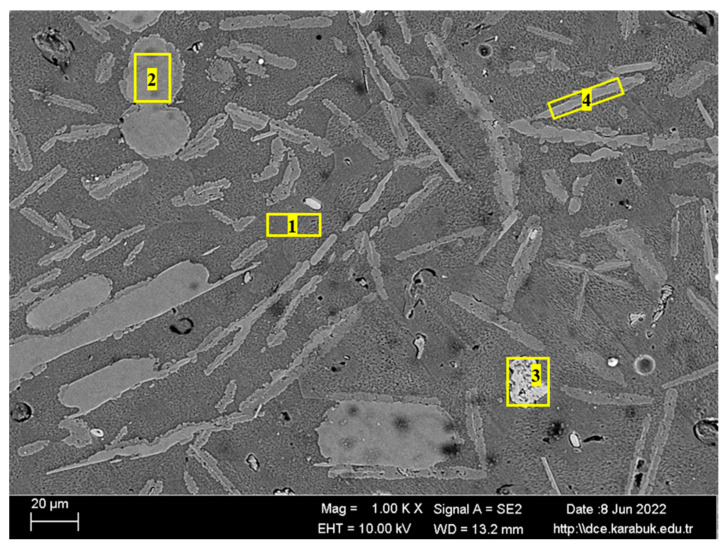
SEM image of the rolled Al7075+8%Ti alloy at 1 k×.

**Figure 6 materials-16-04413-f006:**
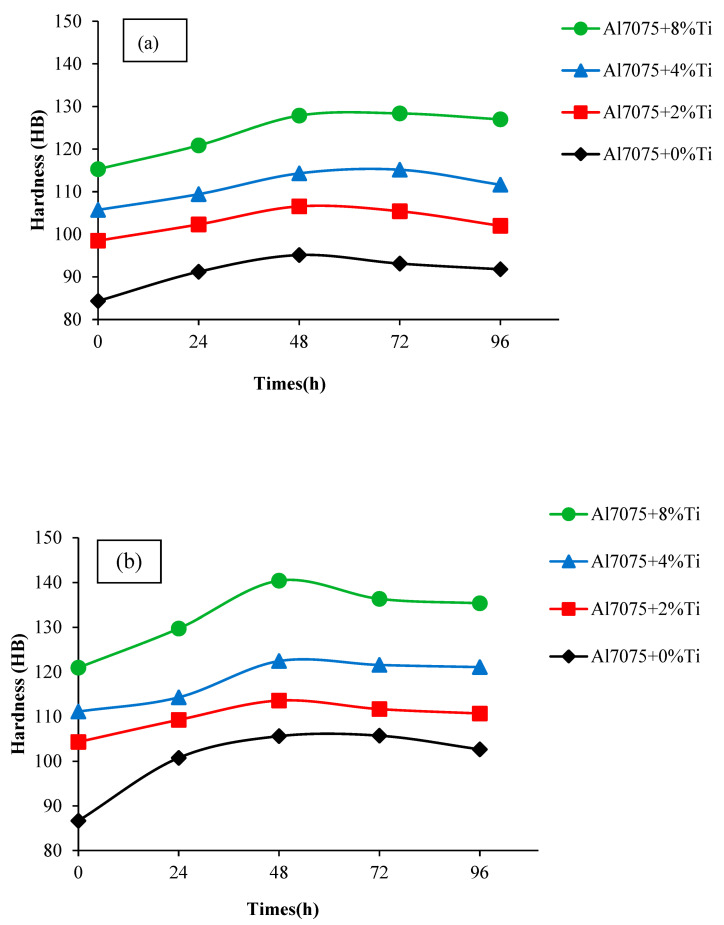
Brinell hardness-measurement results of the (**a**) unrolled and (**b**) rolled Al7075+0%Ti-, Al7075+2%Ti-, Al7075+4%Ti-, and Al7075+8%Ti-reinforced alloys after the T6 aging heat treatment up to 96 h.

**Figure 7 materials-16-04413-f007:**
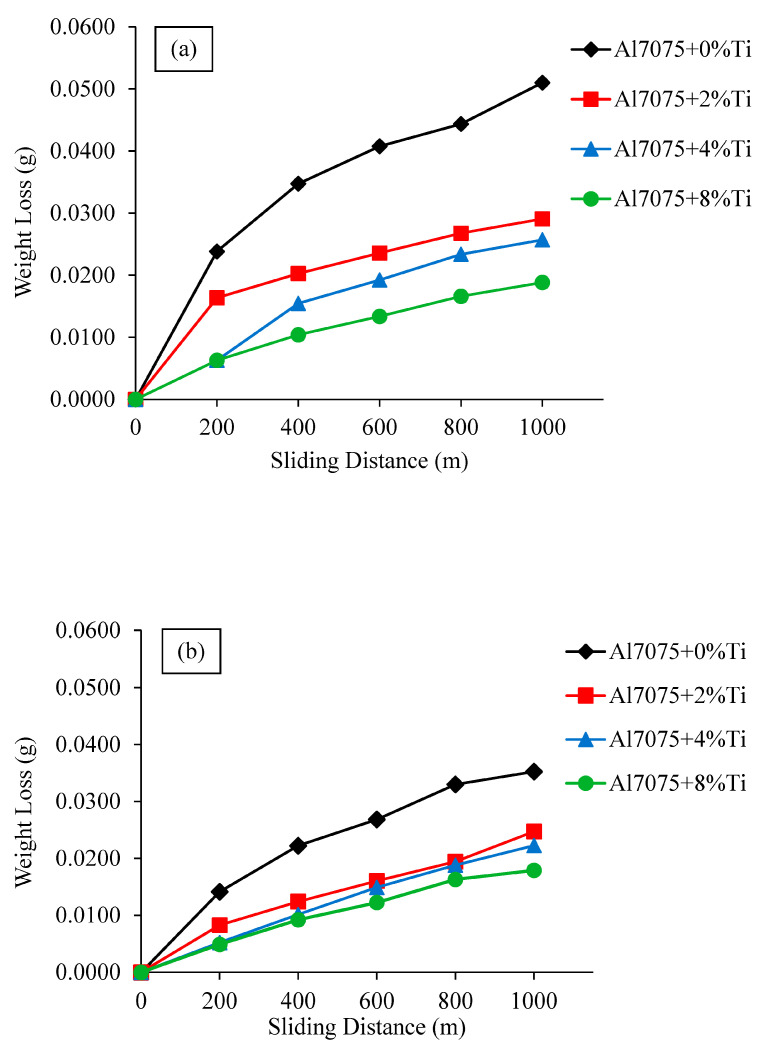
Dry-wear-test results of the (**a**) unrolled and (**b**) rolled Al7075+0%Ti-, Al7075+2%Ti-, Al7075+4%Ti-, and Al7075+8%Ti-reinforced alloys.

**Figure 8 materials-16-04413-f008:**
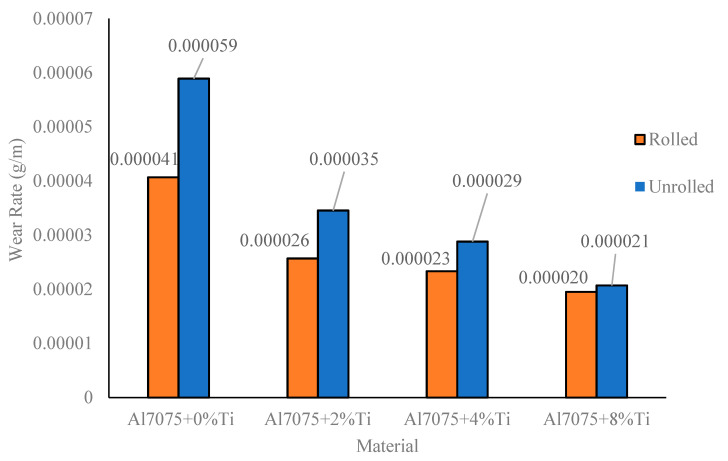
Wear-rate results of the unrolled and rolled Al7075+0%Ti, Al7075+2%Ti, Al7075+4%Ti, and Al7075+8%Ti alloys calculated from the weight-loss versus sliding-distance graphs.

**Figure 9 materials-16-04413-f009:**
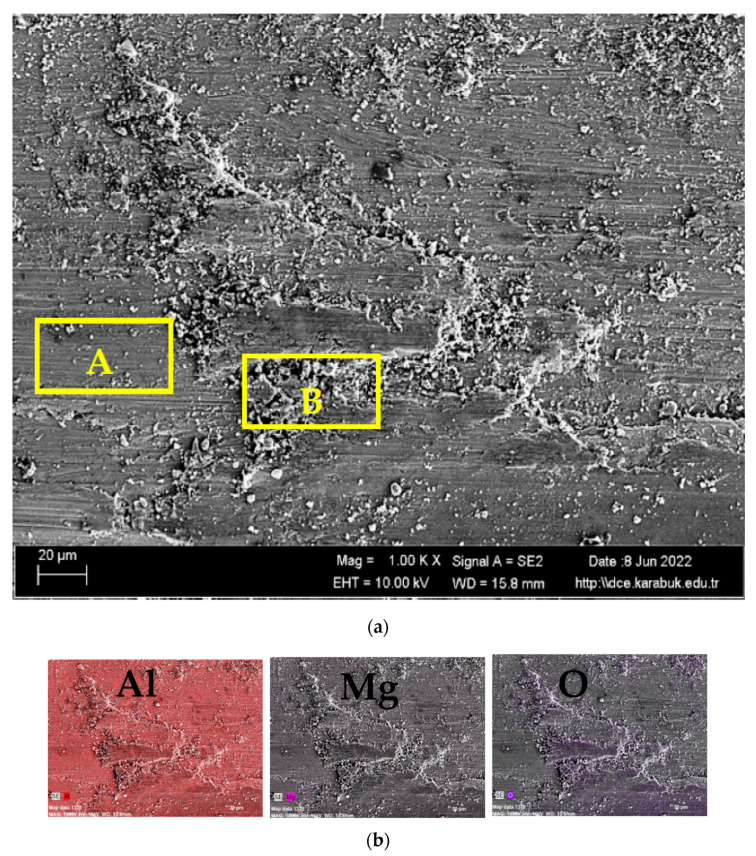
(**a**) SEM image of the worn surface of the unrolled Al7075+0%Ti alloy at 1 k× magnification. (**b**) EDX-MAP analysis results of the elements Al, Mg, and O; marked A shows the abrasive-wear mechanism and the area marked B shows the adhesive-wear mechanism.

**Figure 10 materials-16-04413-f010:**
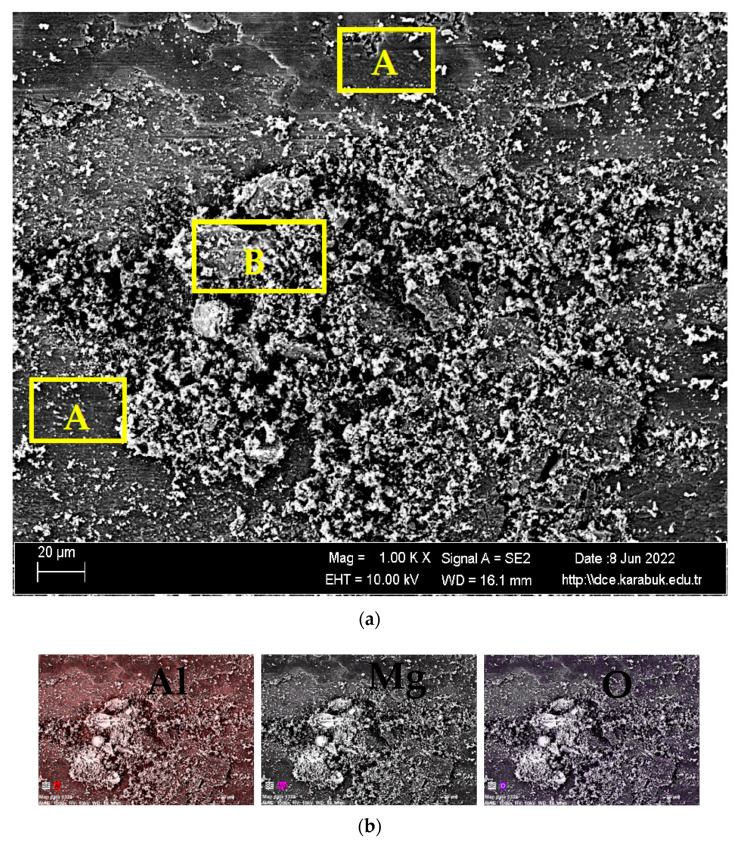
(**a**) SEM image of the worn surface of the rolled Al7075+0%Ti alloy at 1 k× magnification. (**b**) EDX-MAP analysis results of the elements Al, Mg, and O; marked A shows the abrasive-wear mechanism and the area marked B shows the adhesive-wear mechanism.

**Figure 11 materials-16-04413-f011:**
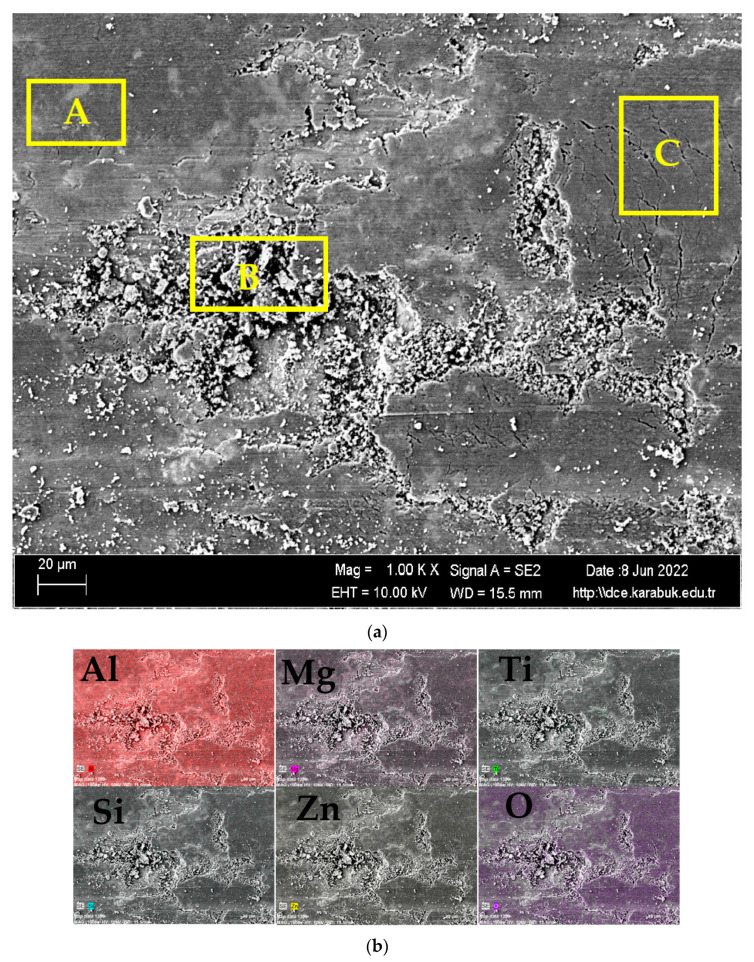
(**a**) SEM image of the worn surface of the unrolled Al7075+8%Ti alloy at 1 k× magnification. (**b**) EDX-MAP analysis results of the elements Al, Mg, Ti, Si, Zn, and O; marked A shows the abrasive-wear mechanism and the area marked B shows the adhesive-wear mechanism, and marked C shows transition region.

**Figure 12 materials-16-04413-f012:**
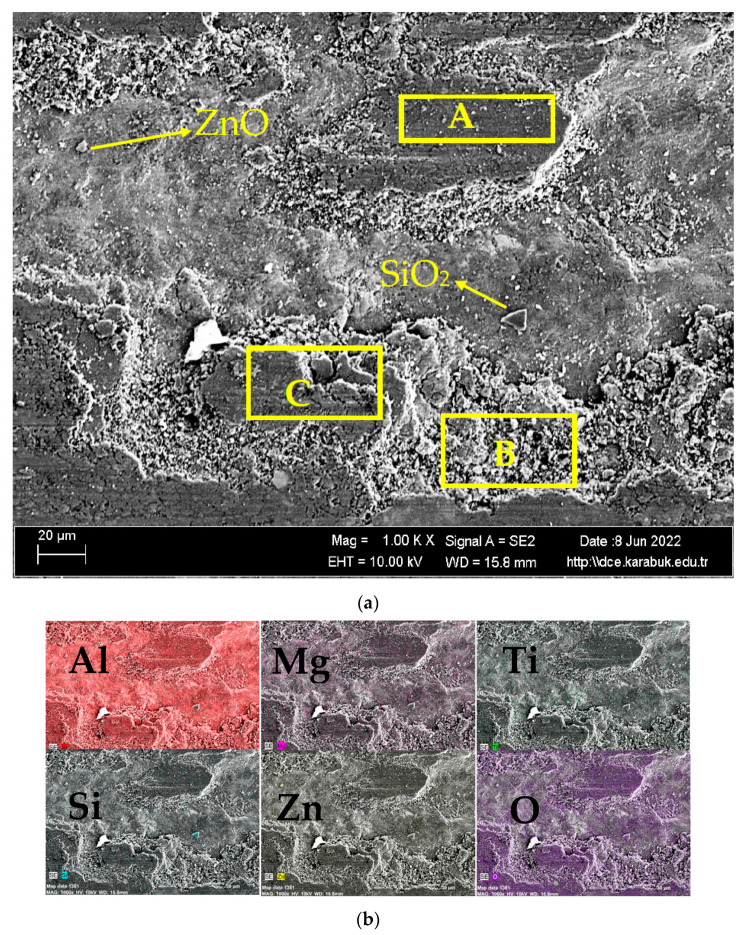
(**a**) SEM image of the worn surface of rolled Al7075+8%Ti alloy at 1 k× magnification. (**b**) EDX-MAP analysis results of the elements Al, Mg, Ti, Si, Zn, and O; marked A shows the abrasive-wear mechanism and the area marked B shows the adhesive-wear mechanism, and marked C shows transition region.

**Figure 13 materials-16-04413-f013:**
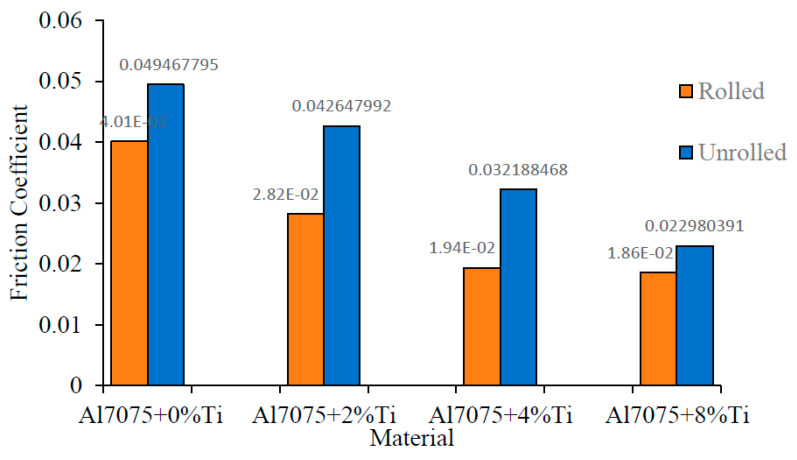
The comparison of the friction-coefficient values of the rolled and unrolled alloys.

**Table 1 materials-16-04413-t001:** The weight contents of the Al7075+0–8%Ti alloys as grams for 1 kg.

Examined Alloys	Weight (g)			
	Al7075	Al-10%Ti Master Alloy	Ti	Mg
Al7075+0%Ti	990.1	0	0	9.9
Al7075+2%Ti	790.2	200	20	9.8
Al7075+4%Ti	590.4	400	40	9.6
Al7075+8%Ti	190.2	800	80	9.2

**Table 2 materials-16-04413-t002:** The chemical compositions of the newly produced alloys.

Chemical Composition	Si	Fe	Cu	Mn	Mg	Cr	Zn	Ti	Al
Al7075+0%Ti	0.4	0.5	1.6	0.3	2.5	0.15	5.5	0.2	Bal
Al7075+2%Ti	0.3	0.39	1.26	0.24	2.5	0.12	4.34	2.2	Bal
Al7075+4%Ti	0.24	0.3	0.96	0.18	2.5	0.09	3.3	4.2	Bal
Al7075+8%Ti	0.08	0.1	0.32	0.06	2.5	0.03	1.1	8.2	Bal

**Table 3 materials-16-04413-t003:** The spectral (EDX) analysis of the selected regions in Figure 3 for the rolled Al7075+0%Ti alloy.

Spectrum	Mg	Al	Si	Ti	Fe	Cu	Zn
1	2.91	85.13	0.07	0.00	0.00	4.21	7.69
2	2.87	85.42	0.13	0.00	0.00	3.54	8.04
3	9.13	61.55	1.07	0.00	0.48	12.16	15.61

**Table 4 materials-16-04413-t004:** The spectral-analysis (EDX) results of the selected regions in Figure 4 for the rolled Al7075+8%Ti.

Spectrum	Mg	Al	Si	Ti	Fe	Cu	Zn
1	1.79	88.74	0.00	0.28	0.00	6.54	2.65
2	0.43	64.24	0.08	35.20	0.00	0.00	0.05
3	0.01	11.27	0.14	0.00	2.62	85.96	0.00
4	2.33	65.65	0.28	31.12	0.00	0.24	0.37

## Data Availability

Not applicable.

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
