# Peer review of "Effect of Aging Heat Treatment on Wear Behavior and Microstructure Characterization of Newly Developed Al7075+Ti Alloys"

_materials, 2023, doi:10.3390/ma16124413_

Round 1

Reviewer 1 Report

This manuscript deals with the tribological behavior assessment of AA7075 alloy with the variable percentage of Ti and further performing the heat treatment.

1-If the heat treatment including T6 and Solution treatment is a part of the study, the title must depict this.

2-The abstract must be improved by including the heat-treatment behavior on wear assessment along with the hardness.

3-Abstrast must highlight the obtained behavior, while the conclusion includes the specific quantitative result, improve the abstract in this manner.

4- This study used the liquid melting process instead of the casting process, how this process is different from the casting, and what are the other processes available? Kindly include it in the introduction.

5- While the authors write they used the casting method for alloy preparation in Section 2, kindly correct if any mistakes.

6-Kindly include the actual pictures of prepared specimens, along with the tribometer setup.

7- Thee explanation and justification for Section 3.1 is quite weak, and require improvement.

8- From section 3.2, does grain size has any relation with the wear resistance? Kindly elaborate.

9- What is the effect of heating time of the observed results apart from the titanium percentage? found any significance?

10- What is gr in the weight loss graph? or its grams gm.

11- Kindly check Figure 7b legends.

12- Kindly check the unit of wear rate in Figure 8.

13- What is the strategy of executing the wear testing, like its the pin on disc, ball of disc? kindly write under section 2 along the wear track path generated.

14-Authors need to include the recently published articles in the references to justify the current significance with the recent trends.

minor and fine-tuning required.

Author Response

Response to Reviewer 1

All of the reviewer's suggestions have been carefully evaluated, and the following changes have been made to the text:

Reviewer 2 Report

The authors investigated the effect of Ti contents on wear behaviour and microstructure of Al7075/Ti composites. I have some comments and suggestions to improve the quality of the current manuscript as below:

1. The author mentioned "newly developed Al7075+Ti alloys" is not true. This alloy has been fabricated and presented in the literature. Please correct it.

2. Abstract should be rewritten (shorter and more concise). Some information about experimental conditions such as "Dry wear tests of all alloys were carried out at a total sliding distance of 1000 m, at a sliding speed of 0.1 m/s and under a load of 20 N" should be only provided in Experimental section.

3. The experimental is not well-designed. According to Table 1 and Table 2, all compositions of Alloy were changed, not only Ti. Therefore, the effect of Ti contents on the properties of the composites is not clear.

4. Microstructure of starting powder (Ti, Al7075, Mg,) should be provided.

5. Density of the prepared alloys should be measured and presented.

6. EDX mapping images presented on Fig 9. Fig 10, Fig 11 and Fig 12 are low quality. They should be revised with better resolution.

7. The method used to calculated the wear rates of the alloys should be provided in Experimental section.

 8. Deeper comments and discussion about XRD patterns of the samples are required for better understanding the phase transformations.

 Moderate editing of English language required.

Author Response

Response to Reviewer

All of the reviewer's suggestions have been carefully evaluated, and the following changes have been made to the text:

Reviewer 3 Report

After thorough study of the manuscript «Wear behaviour and microstructure characterization of newly 2 developed Al7075+Ti alloys» I would recommend to accept it after answering some important questions/notes.

1. Introduction

- Line 46

«Al-Ti alloys are considered as one of the most important material groups of the future because they have the advantages of high elongation ability».

A reference is needed. It is unevident, that addition of Ti can improve ductility.

- Line 82

«Rong et al. [17] studied the impact of the chemical mixture K2TiF6- CaF2-LiCl added to the Al6351 alloy at 720 °C on the sizes and shapes of intermetallic compounds such as Mg2Si, Al3Ti and Al3TiSi0.22».

It seems that this statement doesn’t fit the subject of the paper.

- It would be better not only to make a list of similar studies in the Introduction, but also to demonstrate the potential place your work will occupy among them. For example, what problems remain unresolved in other works. What prompted to expand the concentration interval to 2-8% Ti, if there was already paper, devoted to alloy with 2% Ti.

- Introduction contains a number of references to the studies of Al7075-TiC/TiB2 like composites. Is the comparison with Ti-doped Al7075 alloy appropriate? What is the proposed strengthening mechanism upon the addition of pure titanium? Will it be precipitation strengthening with fine intermetallic particles or solid solution strengthening?

2. Line 153. What is the purpose of Т6 aging?

3. There is a lack of methods and equipment information for XRD, SEM, EDX.

4. Figure 2. XRD patterns should be moved to one graph for space considerations.

5. Line 182. Figure 2 d can not provide information about morphology.

6. Figure 2. What is the reason for the appearance and increase in the intensity of peaks at angles of approximately 35-42 degrees with the addition of titanium? According to the marginal notes, the concentrations of phases in which titanium is absent are increasing.

7. Lines 197-202 – The description is excessive. It should be shortened.

8. Line 209. «Nucleation begins around the initial Al3Ti solid particles created during solidification».

This statement should be proved by references or other ways, because there are several other intermetallic compounds formed, that could initiate the solidification. Also this statement is not suitable for all alloys, since a titaniumless alloy is also discussed, where other intermetallic compounds are present.

9. Line 238. Probably, it means Figure 4.

10. Figure 5. Area 3. Chemical composition of the area doesn’t correspond to any present phase. Probably it is a mixing defect?

11. A short paragraph with conclusions on how titanium modifies the microstructure is needed in the end of Section 3.2.

12. Figure 6. What can be the reason for the maximum hardness at aging time of 48 hours?

13. Line 333. The increase in wear resistance with Ti addition can be attributed to the formation of an oxide film on the surface of the alloy with the heat generated during friction. But the oxide layer based on Al2O3 is expected for every alloy?

14. What is the reason for changing of wear mechanisms for rolled and unrolled conditions?

15. Figure 11b doesn’t give an opportunity to evaluate the distribution of titanium and, accordingly, its effect on the wear mechanism.

16. In the Conclusions the increase in wear resistance is associated only with an increase in hardness and also an increase in oxidation is mentioned. Perhaps, as one of the conclusions, it should be said about the decrease in the coefficient of friction.

Author Response

(The authors gave the same response as above.)

Round 2

Reviewer 2 Report

The manuscript could be accepted.

Minor editing of English language required

Reviewer 3 Report

Thank you for answering the comments.